

# First forcing estimates from the future CMIP6 scenarios of anthropogenic aerosol optical properties and an associated Twomey effect

Stephanie Fiedler[1], Bjorn Stevens[1], Matthew Gidden[2], Steven J. Smith[3], Keywan Riahi[2], and Detlef van Vuuren[4,5]

[1]Max Planck Institute for Meteorology, Bundesstrasse 53, 20146, Hamburg, Germany
[2]International Institute for Applied Systems Analysis, Schlossplatz 1, A-2361 Laxenburg, Austria
[3]Joint Global Change Research Institute, Pacific Northwest National Laboratory, College Park, MD, USA
[4]Utrecht University, Vening Meineszgebouw A, Princetonlaan 8a, 3584 CB Utrecht, The Netherlands
[5]PBL Netherlands Environmental Assessment Agency, The Hague, The Netherlands

*Correspondence to:* Stephanie Fiedler (stephanie.fiedler@mpimet.mpg.de)

**Abstract.** We present the first forcing interpretation of the future anthropogenic aerosol scenarios of CMIP6 with the simple plumes parameterisation MACv2-SP. The nine scenarios for 2015 to 2100 are based on $SO_2$ and $NH_3$ emissions for use in CMIP6 (Riahi et al., 2017; Gidden et al., in prep.). We use the emissions to scale the observationally informed anthropogenic aerosol optical properties and the associated effect on the cloud albedo of present-day (Fiedler et al., 2017; Stevens et al., 2017) into the future. The resulting scenarios in MACv2-SP are then ranked according to their strength in forcing magnitude and spatial asymmetries. Almost all scenarios show a decrease in anthropogenic aerosol by 2100 with a range of 108% to 36% of the anthropogenic aerosol optical depth in 2015. We estimate the spread in the radiative forcing associated with the scenarios in the mid-2090s by performing ensembles of simulations with the atmosphere-only configuration of MPI-ESM1.2. MACv2-SP herein translates the CMIP6 emission scenarios for inducing aerosol forcing. With the implementation in our model, we obtain forcing estimates for both the shortwave instantaneous (RF) and effective radiative forcing (ERF) relative to 1850. Here, ERF accounts for rapid atmospheric adjustments and natural variability internal to the model. The spread for the mid-2090s is -0.20 to -0.57 $Wm^{-2}$ (-0.15 to -0.54 $Wm^{-2}$) for RF (ERF) of anthropogenic aerosol, associated with uncertainty in the emission pathway alone, i.e., the mid-2090s forcing ranges from 33-95% (30-108%) of the mid-2000s RF (ERF). We find a larger ERF spread of -0.15 to -0.92 $Wm^{-2}$, when we additionally consider uncertainty in the magnitude of the Twomey effect. The year-to-year standard deviations around 0.3 $Wm^{-2}$ associated with natural variability highlights the necessity for averaging over sufficiently long time periods for estimating ERF, in contrast to RF that is typically well constrained after simulating just one year. The scenario interpretation of MACv2-SP will be used within the framework of CMIP6 and other cutting-edge scientific endeavours.



## 1 Introduction

Projections of future climate change require plausible assumptions on socio-economic pathways. The sixth phase of the Coupled Model Inter-comparison Project (CMIP6, Eyring et al., 2016) uses the socio-economic pathways described in O'Neill et al. (2014) and quantified by Riahi et al. (2017). Nine different emission scenarios have been defined for CMIP6 and are

described in the framework of the Scenario Model Inter-comparison Project (ScenarioMIP, O'Neill et al., 2016). These emissions have been harmonised and downscaled by Gidden et al. (in prep.). The scenarios include projections of the anthropogenic aerosol emissions that we interpret here with the simple plumes parameterisation MACv2-SP (Fiedler et al., 2017; Stevens et al., 2017).

MACv2-SP parameterises radiative effects of anthropogenic aerosol, e.g., in the atmospheric component of Earth system
models at the Max-Planck-Institute for Meteorology (Giorgetta et al., 2018; Müller et al., 2018; Mauritsen and et al., submitted). Some MIPs endorsed by CMIP6 and other projects require the MACv2-SP parameterisation to perform the requested simulations. Such endeavours are motivated by the benefit of a consistent treatment of aerosol forcing when exploring origins of model differences in radiative forcing (RFMIP, Pincus et al., 2016) and a computationally cheap representation of aerosol effects on climate both in high-resolution simulations (HighResMIP, Haarsma et al., 2016) and in decadal climate predictions
(Miklip and DCPP, Boer et al., 2016; Marotzke et al., 2016). The MACv2-SP parameterization induces aerosol-radiation interaction by prescribing anthropogenic aerosol optical properties and aerosol-cloud interaction in form of a Twomey effect by perturbing the cloud droplet number concentration. The historical development of the anthropogenic aerosol optical depth, $\tau$, has been derived by scaling with the anthropogenic aerosol emission of the past (Stevens et al., 2017).

In the present article, we scale MACv2-SP's $\tau$ in the period $2015-2100$ with the gridded CMIP6 emission scenarios of
$SO_2$ and $NH_3$. The resulting scenarios in MACv2-SP are compared and classified by categories, describing the strength in forcing magnitude and spatial asymmetries. Based on the extremes in projected $\tau$, we present the first CMIP6 estimate of the spread in the radiative forcing of anthropogenic aerosol for the mid-2090s, using the atmosphere component of MPI-ESM1.2 (Mauritsen and et al., submitted). The scaling method and the MPI-ESM1.2 experiments are described in Section 2, followed by the derived temporal developments of $\tau$, the scenarios classification, and the associated spread in aerosol radiative forcing
from MACv2-SP.

## 2 Method

### 2.1 MACv2-SP parameterisation

MACv2-SP is the simple plumes parameterisation for anthropogenic aerosol optical properties and an associated effect on clouds (Fiedler et al., 2017; Stevens et al., 2017). It is informed by the optical properties from the aerosol climatology of
the Max-Planck-Institute for Meteorology version two, MACv2, which includes additional observational data and improved regional corrections (Kinne et al., 2013; Kinne, submitted). For the historical period, MACv2-SP uses the emission inventory





endorsed by CMIP6. A detailed technical description of MACv2-SP is given by Stevens et al. (2017). Here, we focus on the description of the key characteristics and the details of the temporal scaling for the future projections.

MACv2-SP parameterises the optical properties and a relative change in the cloud droplet number concentration associated with anthropogenic aerosol as function of the geographical position, height above ground level, time, and wavelength. It is designed for the implementation in a model's radiative transfer calculation. The development of MACv2-SP was inspired by

the need for having a computationally in-expensive and transparent representation of anthropogenic aerosol, an approach with easily changeable settings for facilitating experimentation, and a method that is flexible enough for usage in a hierarchy of model complexity and resolution.

For achieving these aims, MACv2-SP approximates the spatio-temporal distribution of $\tau$ at 550 nm from MACv2 with analytical functions. These are a superposition of two rotated Gaussian distributions at each of the nine regional pollution

maxima, distributed across the globe. We make herein a distinction between purely industrially polluted plumes and those that are additionally affected by seasonally active biomass burning. These regions differ by the level of aerosol absorption such that a different single scattering albedo is assigned, namely $\omega_{550nm} = 0.87$ for biomass burning and $\omega_{550nm} = 0.93$ for industrial plumes with the same asymmetry parameter of $\gamma_{550nm} = 0.63$. The vertical distributions of the aerosol extinction are approximated with beta functions, tuned to match the averaged profiles at the centre of the plumes from MACv2. Properties at

wavelengths other than 550 nm are derived with an assumed Ångstrom exponent $\alpha = 2.0$.

## 2.2 Construction of aerosol scenarios in MACv2-SP

MACv2-SP has month-to-month and year-to-year changes in $\tau$. We adopt the same annual cycle as used for the historical reconstruction (Stevens et al., 2017). The year-to-year changes in the future scenarios are derived from the gridded aerosol emissions of anthropogenic sources and open burning specified by the CMIP6 emission scenarios (Gidden et al., in prep.).

We construct time series of $\tau_i$ for each plume $i$ using emission scaling factors, $E_i(t)$ as function of year $t$ of the Gregorian calendar:

$$\tau_i(t) = \tau_i(2005)E_i(t), \qquad (1)$$

where $\tau_i$ in 2005 is the reference value. The temporal scaling, $E_i(t)$, for each of the nine aerosol scenarios in Tab. 1 are required input of the MACv2-SP parameterisation. They are available as MACv2-SP input files in netCDF format in the supplementary

material. The resulting anthropogenic aerosol $\tau_i(t)$ for each scenario are presented in Section 3.1.

We construct $E_i(t)$ from the gridded CMIP6 emissions, $\epsilon$, of the chemical species, $k$:

$$E_i(t) = \frac{\sum_k w_k \left[\epsilon_{ik}(t) - \epsilon_{ik}(1850)\right]}{\sum_k w_k \left[\epsilon_{ik}(2005) - \epsilon_{ik}(1850)\right]}. \qquad (2)$$

We use here the gridded CMIP6 emission scenarios, which is different from the emission scaling with ISO country codes for the historical reconstruction for MACv2-SP. For the scenarios, we average the anthropogenic emissions $\epsilon_{ik}$ in a twenty

by twenty degree box around each plume centre, where $\tau_i(2005)$ is specified and scaled over time. The spatially averaged emission flux for these nine regions exceeds the global mean by a factor of 8.5, i.e., our approach captures the dominant





anthropogenic sources that contribute one third of the total global emissions in 2005. We have tested the reproducibility of the regional evolution of $\tau_i$ by scaling with emissions averaged around the plume centers. Here, we derived $\tau_i$ from $SO_2$ emissions from a pre-existing database adopting the same spatial averaging, and compared the results against the corresponding $\tau_i$ from a transient aerosol-climate simulation of ECHAM-HAM reading the same emissions as boundary data. The comparison showed that using spatial averages of $SO_2$ already gives a good approximation of $\tau_i$ from the complex model.

Here, we use $NH_3$ emissions in addition to $SO_2$ for considering that not all dominant emission changes over time scale with the $SO_2$ development. Both emissions from open burning and otherwise classified anthropogenic sources are taken into account. The weight $w_k$ describes the relative contribution of the species, namely $w_{SO_2}$ = 0.645 and $w_{NH_3}$ = 0.355, motivated by the present-day ratio between sulphate and ammonia forcing (Stevens et al., 2017). We herein assume spatio-temporal changes of these two species as representative for all anthropogenic aerosol emissions and deliberately omit other aerosol species for

consistency with the scaling approach for the historical time period (Stevens et al., 2017). CMIP6 specifies emissions for 2015 and every tenth year from 2020 to 2100 (Gidden et al., in prep.). We derive $E_i(t)$ at the same times and apply a linear interpolation in between. The historical reconstruction ends in 2014 and the scenarios, beginning in 2015, have a slightly larger $\tau$ by 0.0008 compared to 2014. The values of $E_i(t)$ are all positive definite, except for the European plume in 2100 for SSP4-3.4 with a weakly negative value of -0.01. This implies a slight reduction of the total aerosol burden relative to 1850 when

anthropogenic aerosol was already present in Europe associated with the industrial revolution.

The future changes in $\tau$ of the scenarios further scale MACv2-SP's magnitude of the induced Twomey effect. We mimic a Twomey effect by the pre-factor, $\eta_N$, that is parameterised as function of latitude, $\phi$, and longitude, $\lambda$:

$$\eta_N(\phi,\lambda,t) = 1 + \frac{\mathrm{d}N}{N} = \frac{ln\left[1000\ [\tau(\phi,\lambda,t)+\tau_{\mathrm{bg}}(\phi,\lambda,t)]+1\right]}{ln\left[1000\ \tau_{\mathrm{bg}}(\phi,\lambda,t)+1\right]}. \tag{3}$$

Multiplying $\eta_N$ with the cloud droplet number concentration of the host model changes the cloud optical properties with the an-

thropogenic aerosol perturbation. The background aerosol optical depth, $\tau_{\mathrm{bg}}$, is herein an idealised plume-wise approximation consistent with the setting for the historical time period (Fiedler et al., 2017; Stevens et al., 2017):

$$\tau_{\mathrm{bg}}(\phi,\lambda,t) = \tau_{\mathrm{pl}}(\phi,\lambda,t) + \tau_{\mathrm{gl}}. \tag{4}$$

The components $\tau_{\mathrm{pl}}$ refer to a plume-shaped background and $\tau_{\mathrm{gl}}$ to a global constant. In the standard setup of MACv2-SP, $\tau_{\mathrm{gl}}$ is set to 0.02. This background is intended for MACv2-SP's aerosol-cloud interaction only and should not be confused with

a natural aerosol pattern from observations. As emphasised by Stevens et al. (2017), this approach has been adopted so as to allow models to use their own natural aerosol for representing aerosol-radiation interaction, to optionally tune the clear-sky radiation balance of models, and to keep a simple formulation of the Twomey effect for adjusting the magnitude. The Twomey effect (Twomey, 1974) is qualitatively understood, but the magnitude of aerosol-cloud interactions remains uncertain (Bellouin et al., in prep.). Reasons for the difficulties to constrain the magnitude are, for instance, a shortage of suitable observations,

model biases affecting radiative forcing, as well as the co-variability of meteorology and aerosol (e.g., Stevens and Feingold, 2009; Rosenfeld et al., 2014; Bony et al., 2015; Fiedler et al., 2016; Bellouin et al., in prep.). In result, different changes in $N$ with aerosol have been proposed (Quaas et al., 2006; Andreae, 2009; Carslaw et al., 2013; Stevens et al., 2017). In the present





work, we choose the original formulation by Stevens et al. (2017) for having a consistent treatment from pre-industrial to the future projections. Note that stronger aerosol-cloud interactions are also plausible (Stevens, 2015; Bellouin et al., in prep.) and can be represented by MACv2-SP, e.g., by relaxing $\tau_{gl}$ in Eq. 4 like we do in a set of experiments here and elsewhere (Fiedler et al., 2017).

## 2.3 Calculation strategy for spread in aerosol forcing

We perform climate simulations for estimating the anthropogenic aerosol forcing. For doing so, we use the atmosphere-only model configuration of MPI-ESM1.2 (Mauritsen and et al., submitted) and follow the strategy by Fiedler et al. (2017). Natural variability internal to the model affects the effective radiative forcing estimates (Fiedler et al., 2017). For sufficiently accounting for the natural variability, we run six control experiments without any anthropogenic aerosol for the pre-industrial aerosol of 1850 and six experiments with additionally prescribed anthropogenic aerosol from MACv2-SP for the mid-2090s. We herein

choose three projections of $\tau$ for characterising the spread in anthropogenic aerosol forcing, namely the scenarios SSP1-2.6, SSP3-ref, and SSP5-ref that are introduced in Section 3.1. We perform for each scenario three experiments with the standard settings for $\eta_N$ consistent with the historical reconstruction of Stevens et al. (2017). Additionally, we perform another three experiments for each scenario, where we increase $\eta_N$ for quantifying the sensitivity of the forcing spread to uncertainty in the magnitude of the Twomey effect. Here, we follow the method of the LBG experiments in Fiedler et al. (2017) and set

$\tau_{gl} = 0.002$ in the experiments SSP1-2.6-LBG, SSP3-ref-LBG, and SSP5-ref-LBG of the present article. The historical forcing estimate is for the mid-2000s aerosol pattern and identical with the experiment SP in Fiedler et al. (2017). All simulations are performed for the period $2000-2010$ with the same annually-repeating monthly aerosol patterns for sampling natural variability with the same boundary conditions.

We calculate the effective radiative forcing (ERF), the instantaneous radiative forcing (RF), and their difference as net

contribution from rapid adjustments. ERF is determined as annual differences in the top of the atmosphere shortwave radiation balance with and without $\tau$ from MACv2-SP such that we yield 180 annual estimates of ERF for each mid-2090s aerosol pattern. RF is computed online by calling the radiative transfer calculation twice, i.e., once with and once without $\tau$ of MACv2-SP. We therefore have thirty estimates of RF for each of the aerosol patterns. We further estimate the regional forcing efficacy:

$$E(\phi, \lambda, t) = \frac{\mathrm{RF}(\phi, \lambda, t)}{\tau(\phi, \lambda, t)} \tag{5}$$

for assessing the co-variability of RF with $\tau$. All forcing analyses are for $2001-2010$, i.e., without the first year of the simulations due to the model spin-up.

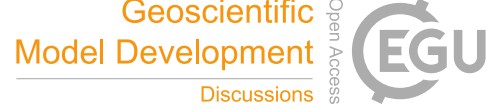

## 3 Results

### 3.1 Scenarios of future anthropogenic AODs

#### 3.1.1 Regionally averaged projections

The results for $\tau_i(t)$ are shown in Fig. 1 for each of the nine CMIP6 scenarios, listed in Tab. 1. Anthropogenic $\tau$ projections for East Asia are decreasing and reach levels comparable to the present-day conditions in Europe in the scenarios of SSP1, SSP2, and SSP4 by the middle of the 21st century or later. The scenarios of SSP3 and SSP5 also show decreasing $\tau$ by the end of the 21st century, but the level first increases and does not reduce as drastically as in the other SSPs. The development for Africa's anthropogenic $\tau$ is typically an increase by 2100. This projection is particularly pronounced in SSP3 and SSP4 with

an increase from 2015 to 2100 by factors around two to four. The scenarios of SSP1 and SSP2 have slight $\tau_i$ increases in Africa only, whereas the scenarios of SSP5 assume an increase in Africa's $\tau$ towards the 2040s and 2080s and a subsequent decrease, respectively. Anthropogenic $\tau$ in Europe and the Americas remain comparable to the low present-day levels.

The analytical functions in MACv2-SP construct the temporally changing $\tau$ patterns from the time series of $\tau_i$. Examples of the resulting spatial distributions of $\tau$ are shown for the mid-2050s and mid-2090s from all nine scenarios in the appendix (Fig.

A1 and A2). We separate in the following MACv2-SP's scenario interpretation into differences in global mean magnitudes and spatial patterns of $\tau$. A summary of the global mean $\tau$ and the scenario categories is provided in Tab. 1.

Magnitude differences are shown as globally averaged $\overline{\tau}$ and $\overline{\eta_N}$ in Fig. 2. SSP3-ref stands out as the high-end scenario for both $\overline{\tau}$ and $\overline{\eta_N}$ with values exceeding the historical reconstruction for the mid-2000s and the mid-1970s. This scenario depicts socio-economic development failures associated with increasing air pollution (Riahi et al., 2017). Strong aerosol forcing is also

expected from SSP4-6.0 and SSP5-ref. In contrast, steep decreases in aerosol forcing are expected in SSP1, although $\tau$ reaches minima around 0.012 only after the 2030s. This scenario reflects the assumption of stringent pollution controls (Riahi et al., 2017). Towards the end of the 21st century, the spread in $\overline{\tau}$ is largest between SSP3-ref and SSP1, with a range of 108% to 36% of the $\overline{\tau}$ in 2015 (Tab. 1). We therefore build our estimate of the spread in forcing for the mid-2090s on these scenarios (Section 3.2).

#### 3.1.2 Spatial $\tau$ asymmetries

For assessing scenario differences in the spatial patterns, we calculate the hemispheric asymmetry from zonally averaged $\overline{\tau}$ for each latitude, $\phi$:

$$A = \frac{\overline{\tau(\phi)} - \overline{\tau(-\phi)}}{2}. \tag{6}$$

In a second step, we divide $A$ by the global mean of the same scenario, $\overline{\tau}$, for screening out magnitude differences. The results

are shown for three years in Fig. 3. We find similarly large $A$ for SSP2-4.5, SSP3-ref, and SSP3-low that are closest to the value for 2015. For all scenarios, $A$ is particularly large in the tropics and sub-tropics, here defined as regions equatorwards of 36° and referred to as low latitudes in the following.





All scenarios project a gradual decrease in the averaged hemispheric $\overline{A}$ in the course of the 21st century. This implies that the zonally averaged $\tau$ becomes increasingly symmetrically distributed about the equator in stark contrast to the historical reconstruction when most anthropogenic aerosol was in the northern hemisphere. We show these temporal evolutions of $\overline{A}$

and also the ratio, $\overline{A}_{\text{low}\phi}/\overline{A}_{\text{high}\phi}$, of the mean $\overline{A}$ in low latitudes relative to the high-latitude mean in Fig. 2. The mean $\overline{A}_{\text{low}\phi}$ exceeds the value of $\overline{A}_{\text{high}\phi}$ by at least a factor of two (Fig. 2), indicating that most $\tau$ is in latitudes equatorwards of 36°. SSP4-3.4 has herein by far the strongest contrast between the low and high latitudes with a factor of roughly 11 in 2100, but the overall smallest hemispheric $A$. This behaviour reflects the relatively symmetric $\tau$ about the equator (Fig. A2). The scenarios of SSP1 also has a stronger concentrations of anthropogenic aerosol in the low latitudes than further polewards, but a moderate

hemispheric $A$ compared to more extreme scenarios. Overall, the scenarios of SSP5 have the smallest differences between the low and high latitudes.

The scenarios in MACv2-SP are constructed with the same scaling values for the annual cycles in $\tau$, shown in Fig. A3. Their main differences in the annual cycle of $\tau$ are associated with the variety in the spatial patterns discussed aloft. Similarities amongst the scenarios are marked (1) tropical $\tau$ maxima in the scenarios of SSP4 and SSP5-ref for June to January, and (2)

northern high-latitude $\tau$ maxima in SSP2, SSP3, and SSP5. The former maxima are herein associated with anthropogenic aerosol from biomass burning, whereas the latter is dominated by industrial emissions.

### 3.2 Radiative forcing of anthropogenic aerosol

#### 3.2.1 Global means

We choose SSP3-ref as high-end scenarios and SSP1-2.6 as a lower bound for the $\overline{\tau}$ spread of 0.009 to 0.027 at the end of the

20 21st century for investigating the associated differences in radiative forcing. We additionally simulate SSP5-ref that also has a high $\overline{\tau}$ of 0.022, but interesting differences in the spatial patterns compared to SSP3-ref. SSP5-ref projects most aerosol in Africa, while SSP3-ref most in Asia. The aerosol in Africa is seasonally dominated by biomass burning, whereas aerosol in East Asia is primarily associated with industrial emissions. SSP3-ref has herein the largest hemispheric $\overline{A}$ (Fig. 3), whereas the annual cycle in SSP5-ref is more strongly pronounced due to the seasonally active biomass burning in Africa (Fig. A3).

Tab. 2 summarises the global mean estimates of RF and ERF. Compared to the mid-2000s, RF and ERF decrease in all scenarios, except in SSP3-ref with the highest aerosol burden. In SSP3-ref, the mid-2090s ERF of -0.54 $\text{Wm}^{-2}$ is namely slightly stronger than for the historical estimate of -0.50 $\text{Wm}^{-2}$ for the mid-2000s from Fiedler et al. (2017). The mid-2090s RF and ERFs are $33-95\,\%$ and $30-108\,\%$ of the mid-2000s estimates from Fiedler et al. (2017). This forcing spread of -0.15 to -0.54 $\text{Wm}^{-2}$ for the mid-2090s describes the uncertainty associated with the future emission pathways alone. When

we additionally consider the uncertainty in the magnitude of the Twomey effect, we get a larger ERF spread of -0.15 to -0.92 $\text{Wm}^{-2}$ (Tab. 2). Our estimates are consistent with earlier studies, namely the scenario spread of -0.7 to -1.0 $\text{Wm}^{-2}$ at the end of the 21st century from the CMIP5 configuration of HadGEM2-ES (Bellouin et al., 2011), the ACCMIP model mean estimate of -0.12 $\text{Wm}^{-2}$ for 2100 (Shindell et al., 2013) and the scenario spread in clear-sky RF of -0.24 to -0.37 $\text{Wm}^{-2}$ for 2100 from Lamarque et al. (2011).





Estimating ERF requires accounting for variability internal to the model. Fig. 5 shows the distribution of yearly estimates of ERF from the ensemble of simulations for the mid-2090s. The year-to-year standard deviations of around $0.3\,\mathrm{Wm^{-2}}$ are herein comparable between the mid-2000s and all the projections for the mid-2090s. This behaviour reflects that a precise estimate of

5 ERF for any given aerosol distribution and strength requires averaging over several decades (Fiedler et al., 2017). Particularly when the ERF is small, e.g., for SSP1-2.6, the year-to-year standard deviation is even larger than ERF itself (Tab. 2). Compared to ERF, the year-to-year standard deviation of RF is small, indicative of comparably stable estimates such that a one-year mean is typically sufficient for a precise estimate of a model's RF. It implies that the model-internal variability in ERF is primarily associated with the variability in the net contribution of rapid adjustments (Fig. 5).

### 3.2.2 Spatial patterns

The regional pattern of the clear-sky contributions to RF resembles the distribution of $\tau$, shown in Fig. 6. Negative radiative effects herein correlate with increasing $\tau$, shown by the similarity of the patterns in forcing efficacy, $E$, for all three scenarios of different forcing strengths. Most regions show the expected negative radiative effects associated with anthropogenic aerosol. The only exception is North Africa, where the more strongly absorbing aerosol at the edge of the biomass burning plume

induces weakly positive radiative effects over the strongly reflective desert surface. In all-sky conditions, the patterns are to a great extent similar to the ones in clear-sky, but clouds mask parts of the negative radiative effects such that the regional all-sky contributions to RF are typically less negative (Fig. 7).

Including rapid adjustments strongly impairs the detectability of significant radiative effects for the mid-2090s aerosol patterns. Fig. 7 and Fig. 8 show the similarly strong impact of atmospheric variability on ERF for both high (SSP3-ref) and

20 low (SSP1-2.6) aerosol forcing categories as well as different strengths of the Twomey effect. The impact of natural variability is consistent with findings for the patterns of the mid-1970s and mid-2000s (Fiedler et al., 2017). An interesting feature in the projection for the mid-2090s is the positive forcing in parts of central Africa. This pattern emerges primarily from rapid adjustments in the atmosphere with a relative smaller regional contribution from RF (Fig. 7).

## 4 Conclusions

The present article presents the MACv2-SP interpretation of the future CMIP6 emissions of anthropogenic aerosol. We show the construction of the scaling parameter for the aerosol optical depth, $\tau$, for $2015-2100$ and the resulting spatio-temporal distribution of $\tau$. The highlights of the projected aerosol developments for the 21st century are (1) a continuous stabilisation or further decrease in $\tau$ in Europe and the Americas, (2) a long-time decrease of $\tau$ in East Asia stretching over the next decades in many scenarios, and (3) steep increases in $\tau$ in Africa's biomass burning regions in most scenarios. We rank the scenarios with

respect to their strengths in the aerosol forcing magnitude, the hemispheric asymmetry and the low- to high-latitude asymmetry, summarised in Fig. 9.

We estimate the spread in radiative forcing of anthropogenic aerosol at the end of the 21st century that is associated with uncertainty in future aerosol emissions (Fig. 10). For doing so, we choose three aerosol forcing scenario (SSP5-ref, SSP3-ref,



and SSP1-2.6). Their MACv2-SP aerosol is prescribed in ensembles of simulations with the atmosphere-only configuration of MPI-ESM1.2 (Mauritsen and et al., submitted), participating in CMIP6 and endorsed MIPs. The ensemble is herein useful for estimating the effective radiative forcing in light of natural variability internal to models (Fiedler et al., 2017). The year-to-year

standard deviation in ERF of roughly $0.3\,\mathrm{Wm}^{-2}$ illustrates the impact of natural variability on ERF estimates that almost exclusively stems from the variability in the net contribution from rapid adjustments. Averaging over sufficiently long time periods, here 180 years, accounts for that variability. MPI-ESM1.2 gives a spread of -0.15 to $-0.54\,\mathrm{Wm}^{-2}$ in ERF of anthropogenic aerosol for the mid-2090s (Fig. 10), reflecting the overall uncertainty due to the anthropogenic emission pathways alone. The clear-sky forcing is herein slightly stronger with -0.24 to $-0.69\,\mathrm{Wm}^{-2}$ since the clouds mask radiative effect of anthropogenic

aerosol. Additionally considering uncertainty in the magnitude of the Twomey effect widens the spread in all-sky ERF to -0.15 to $-0.92\,\mathrm{Wm}^{-2}$.

MACv2-SP's interpretation of the CMIP6 emission scenarios will be applied in MIPs endorsed by CMIP6 and other research activities ranging from high-resolution modelling via seasonal and decadal climate predictions to climate-change studies. Past studies have highlighted the role of aerosol radiative forcing for climate changes (e.g., Chung and Soden, 2017) underlining the

15 importance of better understanding the uncertainty of anthropogenic aerosol forcing. Research initiatives such as the radiative forcing model inter-comparison project (RFMIP, Pincus et al., 2016) can help to make progress in understanding model biases causing diversity in aerosol forcing. RFMIP adopts MACv2-SP including the here-presented high-end scenario SSP5-ref. All scenario input files for MACv2-SP are freely available in the supplementary material of this publication. We hope MACv2-SP's scenarios will be useful for advancing our understanding of climate change and supporting impact studies for informing

stakeholders.

*Code and data availability.* The future scaling for MACv2-SP are available in the supplementary material of this article and via input4MIPs. The code and the historical scaling of MACv2-SP is available via input4MIPs, and as supplementary material of Stevens et al. (2017). MPI-ESM1.2's code and the experiment data is stored in the tape archive of DKRZ and accessible by contacting publications@mpimet.mpg.de.

*Author contributions.* SF led the writing of the manuscript, constructed the scaling parameters for MACv2-SP, preformed the climate model

simulations for the forcing calculations, and analysed the data. BS conceived the general concept of MACv2-SP. MG led the ScenarioMIP analysis and data generation effort and provided data for this paper. SJS led efforts to downscale scenario data to grids. KR and DvV led the coordination of ScenarioMIP. All authors contributed to the content of the manuscript.

*Competing interests.* The authors have no competing interests.





*Acknowledgements.* We acknowledge the use of the data from the CMIP6 emission scenarios and the supercomputer facilities of the DKRZ. SF and BS thank the Max-Planck-Society for funding this work.





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





**Figure 1.** Future $\tau_i$ in the MACv2-SP interpretation of CMIP6 scenarios. Shown are the temporal developments of $\tau$ at 550 nm in the colour-coded aerosol plumes of MACv2-SP for the nine emission scenarios of CMIP6. Note the different scales.



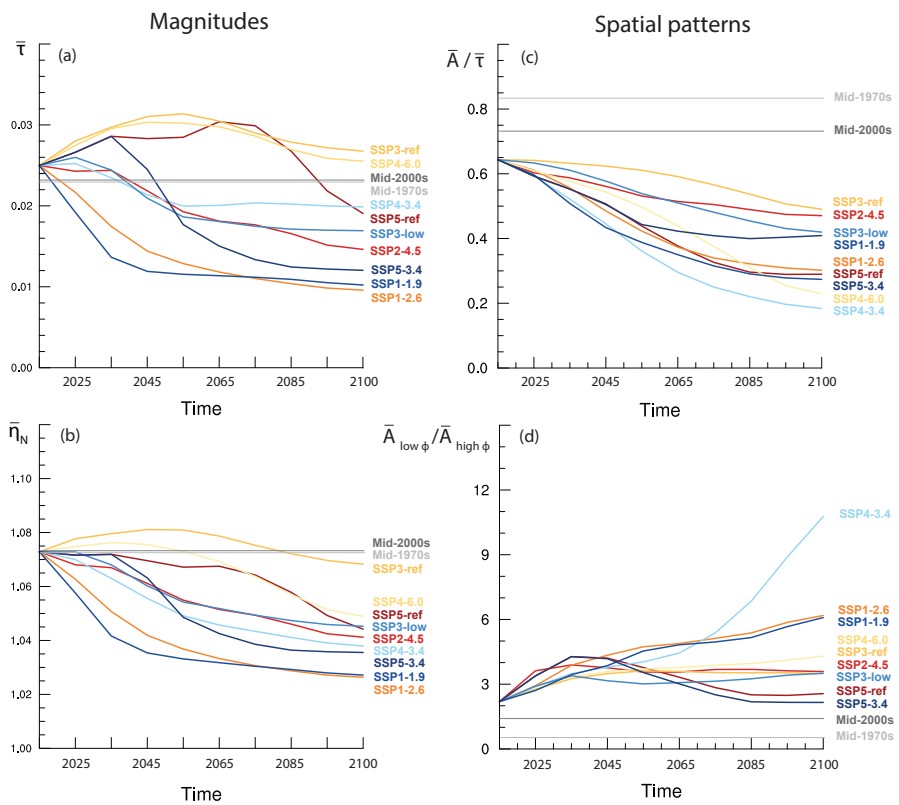

**Figure 2.** Future developments of global means in the MACv2-SP interpretation of CMIP6 scenarios. Shown are the temporal developments of annual averages in (a) $\overline{\tau}$ at 550 nm for inducing aerosol-radiation interaction and (b) $\overline{\eta_N}$ for mimicking aerosol-cloud interaction, as well as (c) hemispheric asymmetry in $\tau$, $\overline{A}$, weighted by the global mean $\overline{\tau}$ and (d) ratio of $A$ in low latitudes ($\phi < 36°$) relative to higher latitudes. MACv2-SP's ranking of the emission scenarios of CMIP6 for their strength in (left) the forcing magnitude and (right) spatial asymmetry is shown with the colour-coded labels on the right. Reference values of the mid-1970s and mid-2000s from the historical reconstruction (Fiedler et al., 2017; Stevens et al., 2017) are indicated by grey lines.



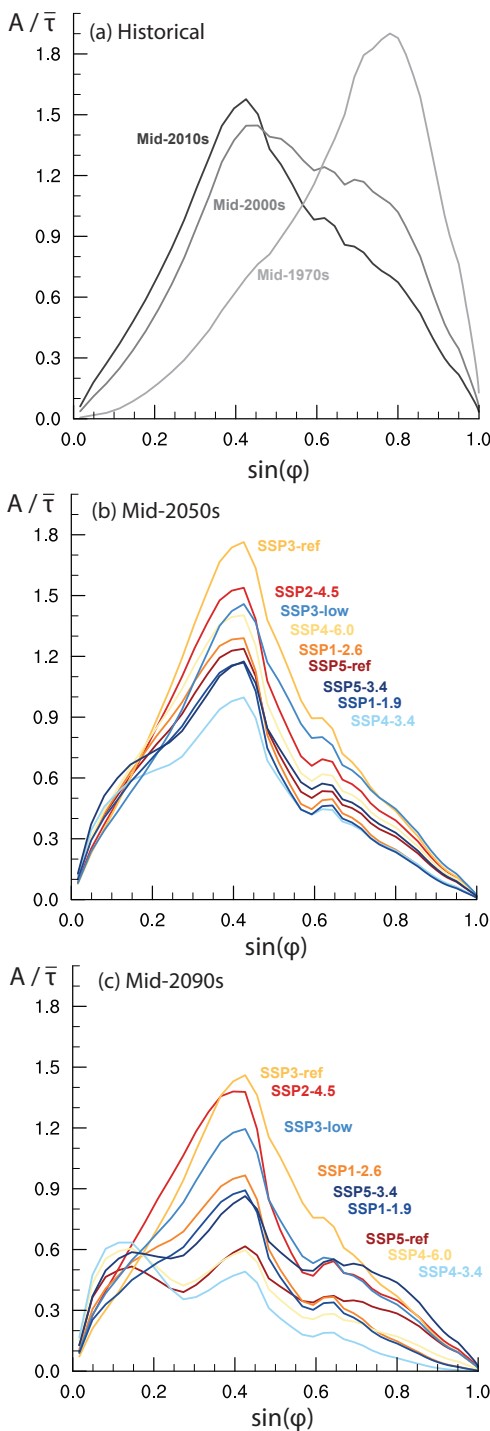

**Figure 3.** Hemispheric asymmetry of $\tau$ in the MACv2-SP interpretation of CMIP6 scenarios. Shown are the hemispheric asymmetry, $A$, weighted by the the global mean $\tau$. All values are computed for $\tau$ at 550 nm for (a) selected years from the historical reconstruction, as well as each CMIP6 emission scenario for both (b) the mid-2050s and (c) mid-2090s.



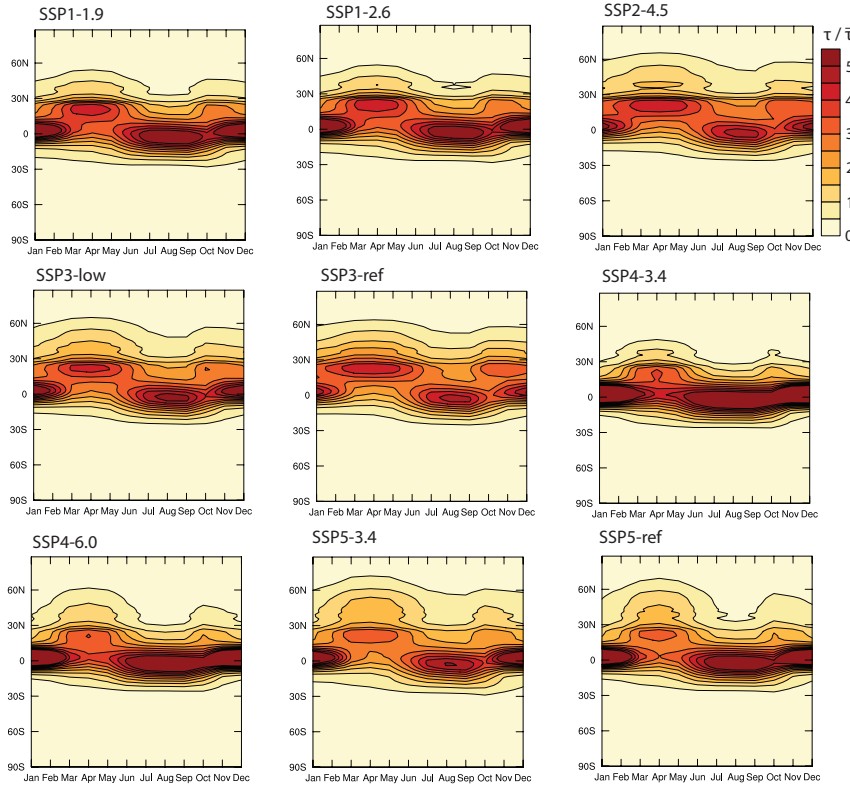

**Figure 4.** Spatial patterns for the mid-2090s in the MACv2-SP interpretation of CMIP6 scenarios. Shown are the annual cycles of the zonal means in $\tau$ at 550 nm weighted by the global mean $\overline{\tau}$ for each month and emission scenario of CMIP6.





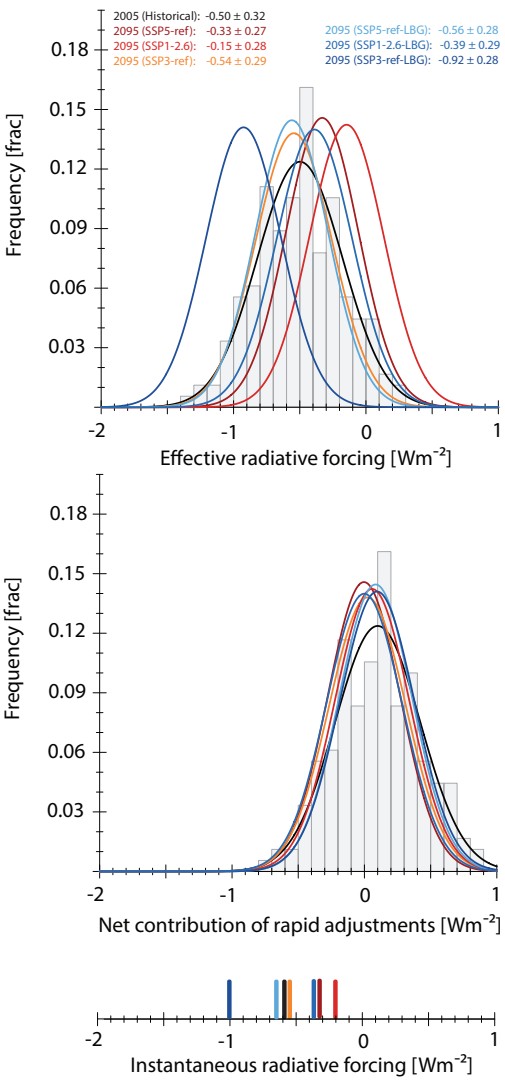

**Figure 5.** Natural variability in all-sky forcing for the mid-2090s. Shown are the Gaussian distributions of (top) the effective radiative forcing (ERF) and (middle) the net contribution of rapid adjustments for the mid-2090s with (orange-red) the standard and (blue) the stronger Twomey effects. The black line marks the mid-2000s values from Fiedler et al. (2017) with the frequency histogram in grey. The distributions are based on annual means for all-sky condition in the shortwave spectrum (SW) at the top of the atmosphere (TOA). The long-term means +/- year-to-year standard deviation of annual ERFs are listed at the top. The year-to-year standard deviation herein illustrate the impact of natural variability internal to the model on estimating ERF, in contrast to the (bottom) annual means in the instantaneous radiative forcing (RF) that are not strongly affected by natural variability.





**Figure 6.** Clear-sky RF and E for the mid-2090s. Shown are the (left) SW TOA instantaneous radiative forcing and (right) forcing efficacy, E, as RF devided by the anthropogenic aerosol optical depth, $\tau$. Contours show $\tau$ at 550 nm from 0.04 in steps of 0.04 (compare Fig. A2). All forcings are for clear-sky conditions in mid-2090s from selected CMIP6 emission scenarios.





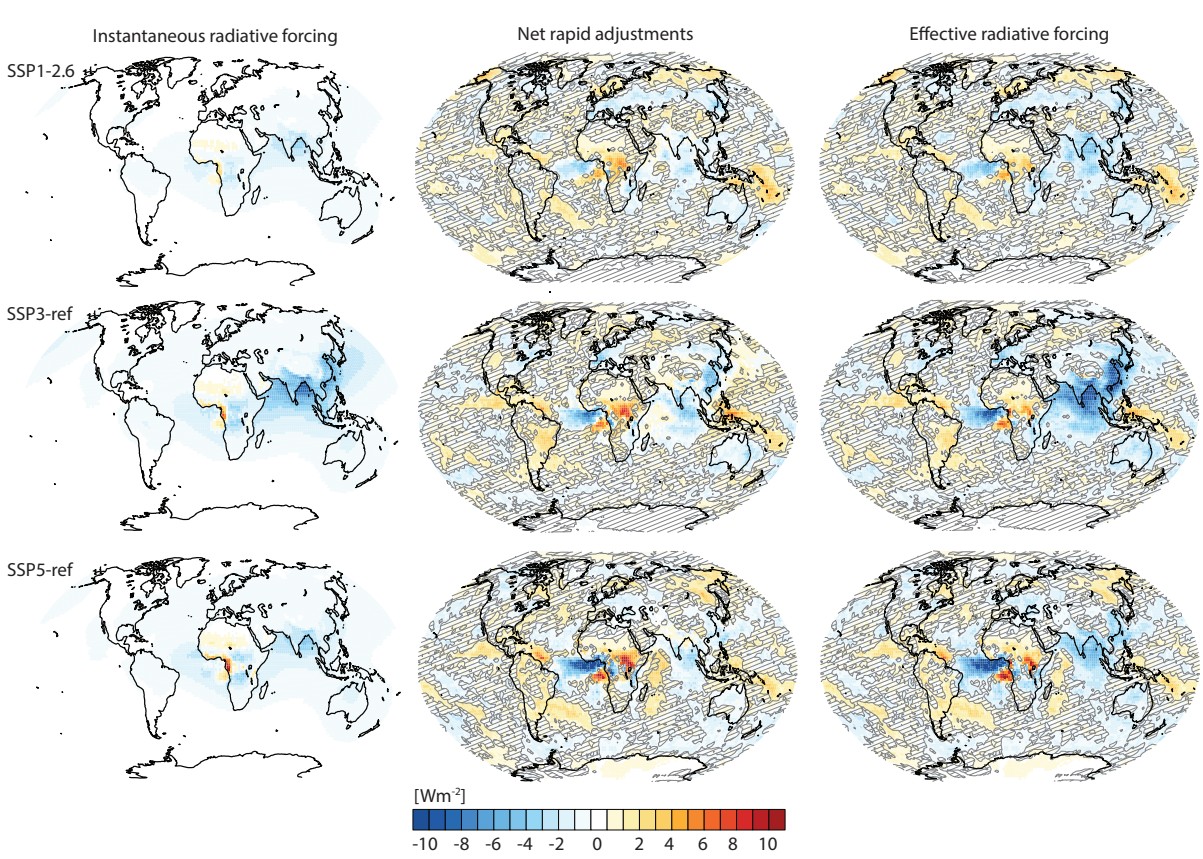

**Figure 7.** All-sky radiative forcing for the mid-2090s. Shown are the SW TOA (left) RF, (middle) the net contribution from rapid adjustments, and (right) ERF for all-sky conditions in the mid-2090s from the CMIP6 emission scenarios. ERFs not significantly different from zero are masked out by hatching, adopting a confidence level of 10%.



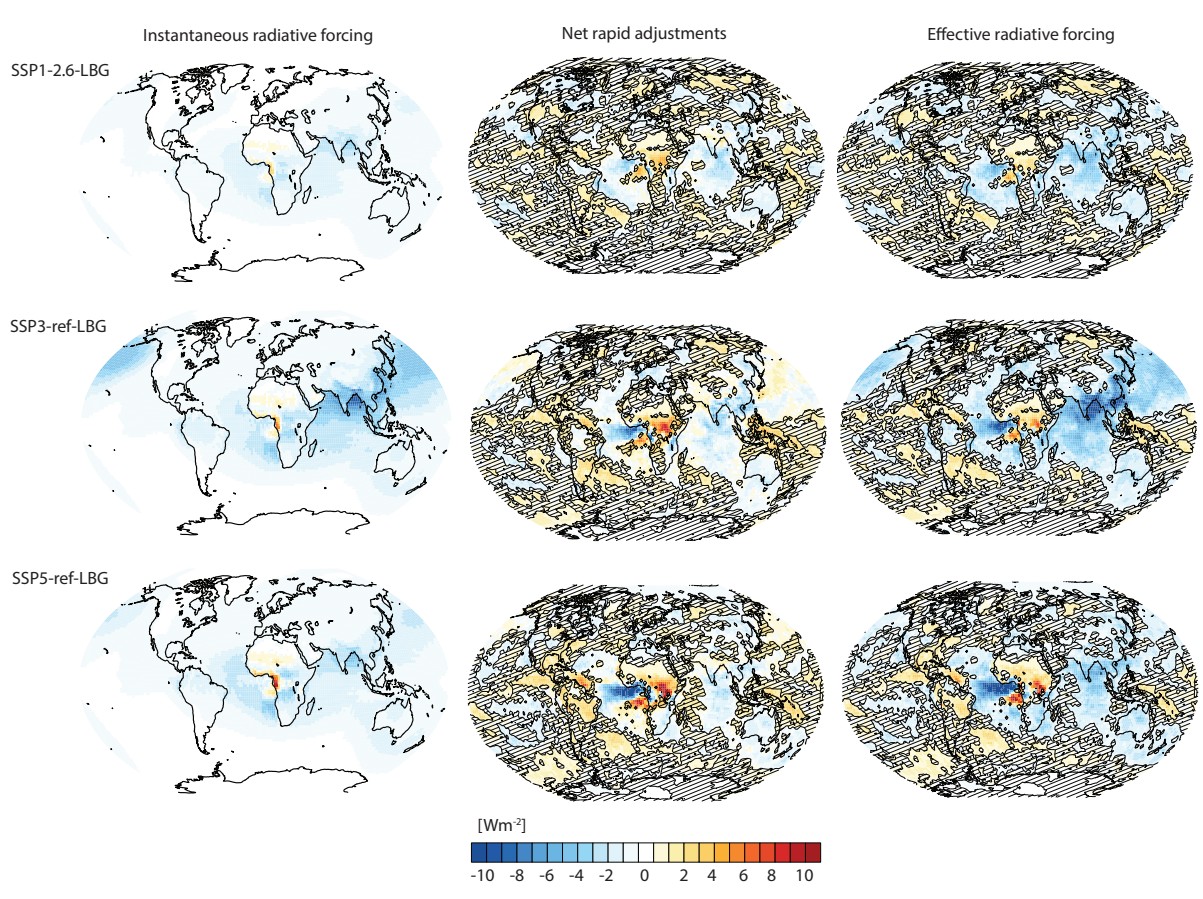

**Figure 8.** All-sky radiative forcing for the mid-2090s with strong Twomey effects. As Fig. 7, but with stronger Twomey effects by increasing $\eta_N$ (Section 2).



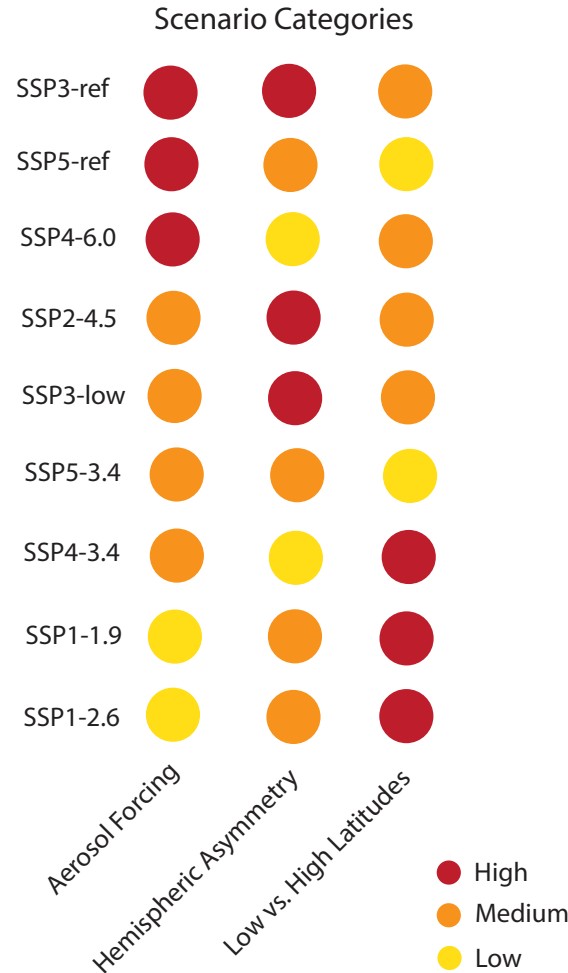

**Figure 9.** Categories in the MACv2-SP interpretation of the CMIP6 anthropogenic aerosol scenarios. Shown is the colour-coded ranking of the CMIP6 scenarios with respect to the strength in the aerosol forcing, the hemispheric asymmetry, and the low- to high-latitude difference in the asymmetry.





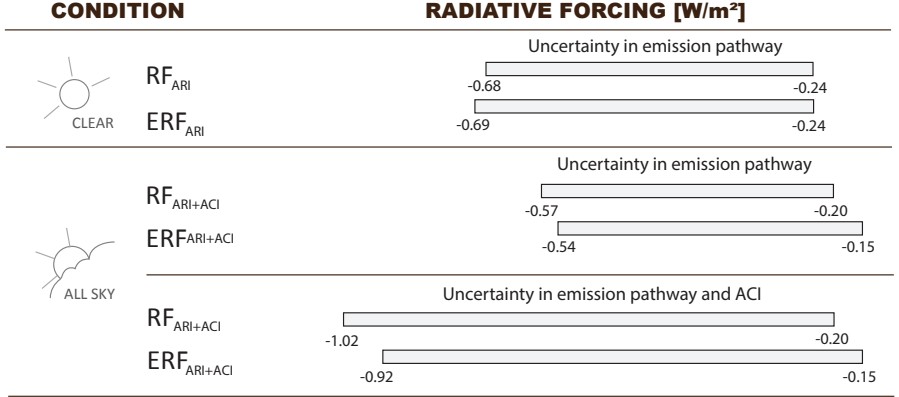

**Figure 10.** Spread in anthropogenic aerosol radiative forcing of the MACv2-SP interpretation of the CMIP6 scenarios for the mid-2090s. Summarised are the instantaneous (RF) and effective radiative forcing (ERF) from aerosol-radiation interaction (ARI) and aerosol-cloud interaction (ACI), based on the high-end scenarios SSP3-ref, SSP5-ref, and SSP1-2.6 as low-emission scenario for the upper bound. The spread is the forcing uncertainty associated with the emission pathways of CMIP6. Additionally, we show the spread when uncertainties in both the emission pathway and the magnitude of the Twomey effect are considered.

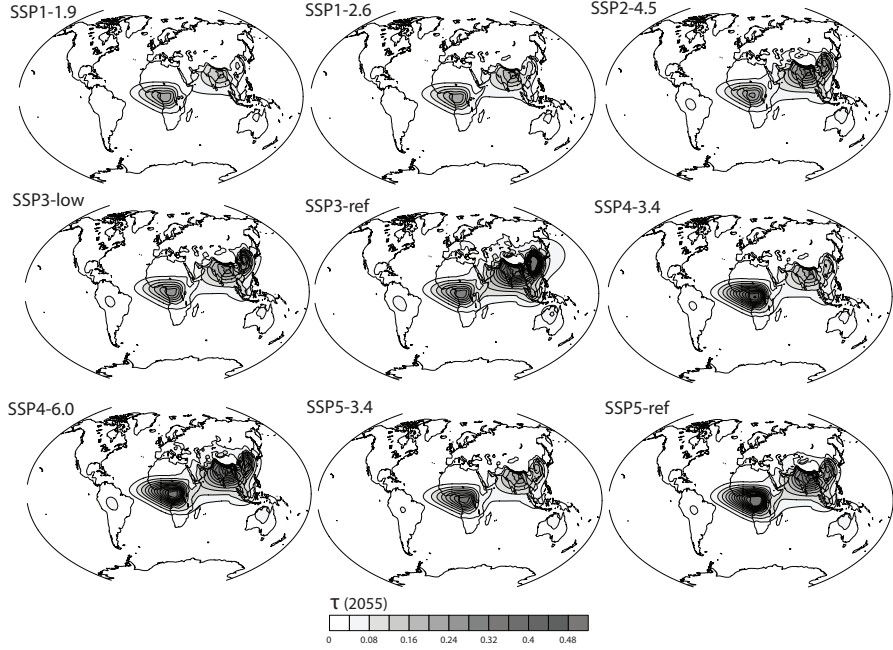

**Figure A1.** $\tau$ scenarios for the mid-2050s in MACv2-SP. Shown are the spatial distributions of $\tau$ at 550 nm in the mid-2050s for each of the nine emission scenarios of CMIP6.





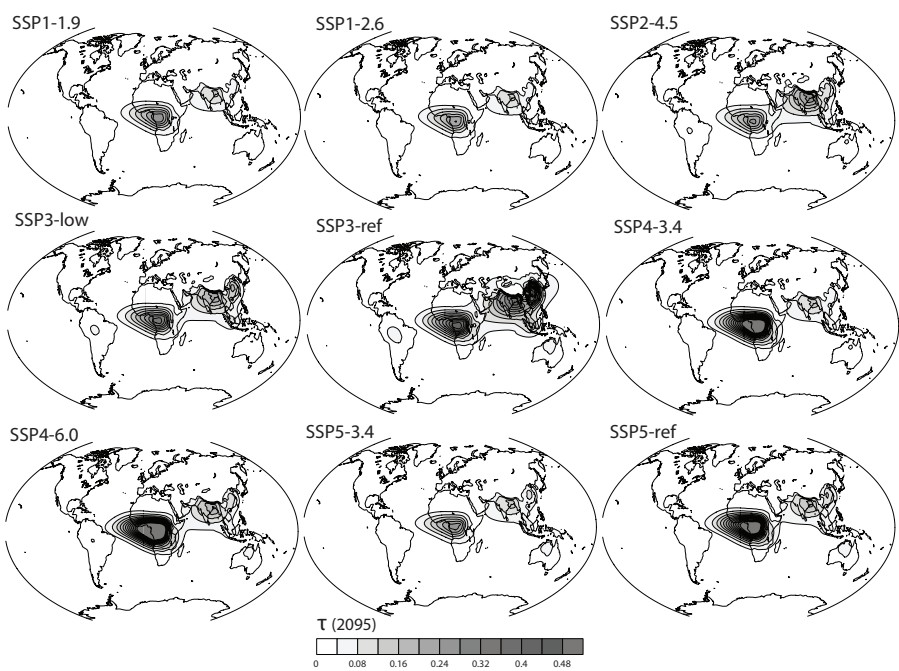

**Figure A2.** $\tau$ scenarios for the mid-2090s in MACv2-SP. As Fig. A1, but projection for the mid-2090s.



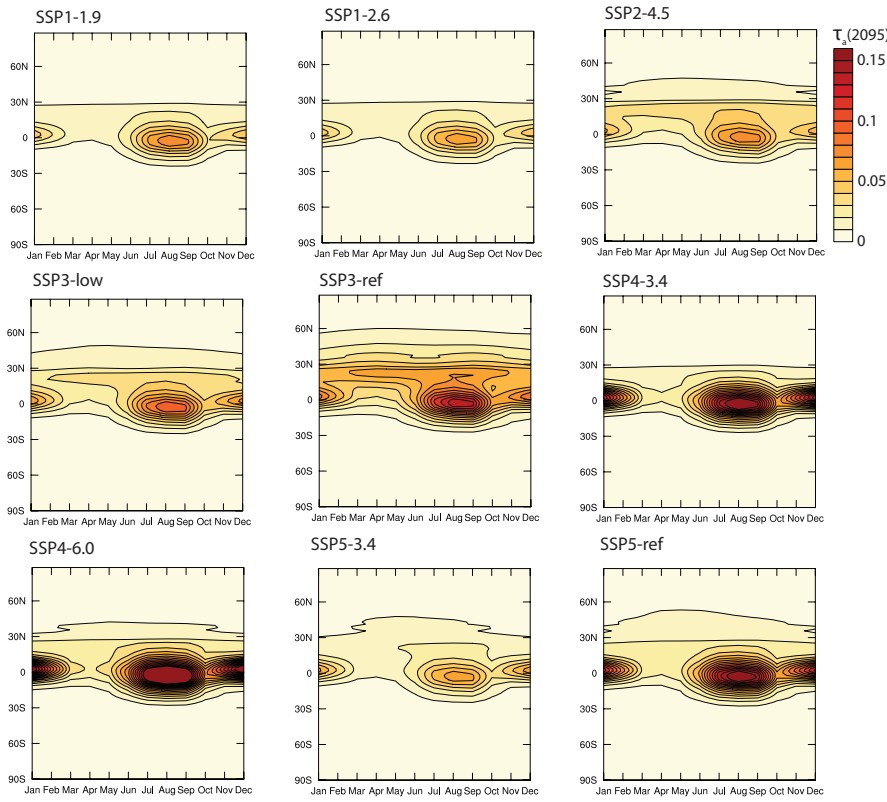

**Figure A3.** $\tau$ scenarios for the mid-2090s in MACv2-SP. Shown are the annual cycles of the zonal means in $\tau$ at 550 nm in the mid-2090s for each of the nine emission scenarios of CMIP6.



**Table 1.** Anthropogenic aerosol scenarios for MACv2-SP ($\overline{\tau}$=0.025 in 2015).

| Scenario name | Mean $\overline{\tau}$ (2050s) | Mean $\overline{\tau}$ (2090s) | Aerosol Forcing category | Hemispheric Asymmetry category | Low vs. High Latitudes category | Usage other than in ScenarioMIP |
|---|---|---|---|---|---|---|
| SSP1-1.9 | 0.012 | 0.011 | Low | Medium | High | |
| SSP1-2.6 | 0.013 | 0.009 | Low | Medium | High | |
| SSP2-4.5 | 0.019 | 0.015 | Medium | High | Medium | DCPP, MiKlip |
| SSP3-low | 0.018 | 0.017 | Medium | High | Medium | |
| SSP3-ref | 0.031 | 0.027 | High | High | Medium | |
| SSP4-3.4 | 0.020 | 0.020 | Medium | Low | High | |
| SSP4-6.0 | 0.030 | 0.026 | High | Low | Medium | |
| SSP5-3.4 | 0.018 | 0.012 | Medium | Medium | Low | |
| SSP5-ref | 0.028 | 0.022 | High | Medium | Low | RFMIP, HighResMIP |

**Table 2.** Mid-2090s anthropogenic aerosol SW (E)RF at TOA from MACv2-SP as long-term averages +/- year-to-year standard deviation in $Wm^{-2}$.

| Year | Scenario | Clear-sky RF | All-sky RF | Clear-sky ERF | All-sky ERF |
|---|---|---|---|---|---|
| mid-2000s | Historical | -0.656 +/- 0.001 | -0.599 +/- 0.003 | -0.67 +/- 0.07 | -0.50 +/- 0.32 |
| mid-2090s | SSP1-2.6 | -0.239 +/- 0.0003 | -0.204 +/- 0.002 | -0.24 +/- 0.07 | -0.15 +/- 0.28 |
| | SSP3-ref | -0.678 +/- 0.001 | -0.568 +/- 0.005 | -0.69 +/- 0.06 | -0.54 +/- 0.29 |
| | SSP5-ref | -0.463 +/- 0.001 | -0.327 +/- 0.006 | -0.45 +/- 0.06 | -0.33 +/- 0.27 |
| mid-2090s | SSP1-2.6-LBG | -0.239 +/- 0.0003 | -0.388 +/- 0.002 | -0.26 +/- 0.07 | -0.39 +/- 0.29 |
| | SSP3-ref-LBG | -0.678 +/- 0.001 | -1.020 +/- 0.006 | -0.69 +/- 0.07 | -0.92 +/- 0.28 |
| | SSP5-ref-LBG | -0.463 +/- 0.001 | -0.644 +/- 0.005 | -0.47 +/- 0.06 | -0.56 +/- 0.28 |