# Peer review of "First forcing estimates from the future CMIP6 scenarios of anthropogenic aerosol optical properties and an associated Twomey effect"

_Geoscientific Model Development, 2018_

## Referee Comment (RC1) · Anonymous Referee #1 · 28 Nov 2018

Fiedler et al. present a modelling study in which they interpret the future emission scenarios of Riahi et al. (2017) using a simple model implemented in a GCM. The two aspects that provide added value compared to the Riahi et al. paper in my opinion are that geographical distributions are shown here, and that the scaling of Stevens et al. (2017) allows to convert the emissions into forcing values given the assumptions in the simple MACv2-SP approach (with some extra model information added from the simulated cloud fraction- and cloud droplet concentration distributions). As far as I understand, one of the co-authors, Gidden, prepares another manuscript for Geophys. Model Devel. that possibly covers the former aspect in a similar way.

[Figure]

The manuscript suffers from poor explanations. It is very difficult to follow the set-up of the simulations (e.g. number of years integrated, boundary conditions for these simulations). The main problem, however, is that it is completely unclear which aerosol species are assessed. In subsection 2.1 of the "Methods" section (p3, l12) the authors explain they distinguish between "biomass burning" and "industrial" aerosol emissions, with substantially different single scattering albedo. This interpretation is taken up later in the discussion of results. In contrast, subsection 2.2 (p4, l7) explains that only sulfate and nitrate are investigated.

In summary, after these explanations are added, and after the specific comments are addressed, I believe the paper has some merit and should be published in Geosci. Model Devel.

p1 l12 – it might be instructive to also report the 2090s forcing relative to 2015.

p2 l10 – Mauritsen et al.: drop "and"

p3 l5 – "need" → "desire"

p3 l27 – w is not explained. Is it some sort of weight? I don't think it is meaningful not to specify the number of species in the sum (and in the text). Isn't it just two?

p3 l29 – p4 l1: I do not understand what the authors do here or mean. A figure might be helpful, or more text to explain.

p4 l4 – is that result shown somewhere, or is there a reference? - the statement needs to be corroborated or withdrawn.

p4 l8 – the relative relevance of sulfate and nitrate depends on the abundance of SO2 and oxidants. It thus seems an oversimplification to assume it is constant with time and geographical location. Some assessment of the error introduced is at least necessary.

p4 l9 – some discussion why in particular absorbing aerosol can be neglected is necessary.
p4 l12 – is this the global mean value (if so, only 0.5% difference?)? Is this similar in individual regions?

p4 l26 – how can this affect clear-sky radiation balance?

p5 l8 – these are equilibrium simulations, if I understand correctly. Then "six" does not mean anything, but the length of the integration is relevant.

p5 l11 – similarly, what does "three experiments" mean in this context?

p5 l13 – more precisely, this is decreasing the tunable parameter \tau_gl, I believe?

p5 l14 – what motivates the name "LBG"?

p5 l17 – what defines a specific year (i.e. why call it "2000 – 2010" rather than "2005 aerosol")? Is the sea surface temperature from observations?

p5 l21 – what is "without \tau", especially for the Twomey effect? Don't the authors rather mean, "with scaling factor (Eq. 2) for 2090 and with scaling factor of 0"? The "180 annual estimates" make me conclude there are 6 realisations (with whatever difference between them) of 30 years integration time each. Is this correct? The authors need to explain clearly what they did.

p5 l23 – I am lost and cannot understand why thirty. What is different between the thirty realisations?

p5 l25 – I strongly suggest not to overload the symbol "E" (that stands for emission scaling) but to use a different one.

p6 l25 – it would be good to motivate this analysis. I would guess that there is no added information here compared to the emission scenarios. Maybe this section can be dropped.

p8 l12 – is this correlation not just by construction of the simple model?

p8 l14 – I did miss the introduction of absorption. Didn't p4 l7 explain that only sulfate

and nitrate are used?

p8 l23 – a discussion on these rapid adjustments is necessary. What exactly happens here in the model?

p12 l1 – journal is missing

Fig. 1 – a figure showing the geographical extent of the plumes is necessary. I suggest to rather use the same scaling for all panels. A global-mean curve would be useful.

Supplementary material: I wasn't able to open the netcdf file. Is there a formatting mistake?

---

## Referee Comment (RC2) · Collins (Referee) · 12 Dec 2018

This paper provides useful documentation of the MACv2-SP aerosols that some models without their own aerosols will use for their CMIP6 simulation.

It should be made very clear in the text that the ERFs from using these aerosols are smaller in magnitude by more than a factor of 2 than assumed by the IAMs when the scenarios were created therefore (all other factors being equal) future temperatures are likely to be less warm in these scenarios (except SSP3-ref) than expected by the IAMs, or compared to interactive aerosol models whose aerosol ERFs are closer to that of the IAMs.

[Figure]

Specific comments:

Page 1, line 2: It is stated that the scenarios are based on SO2 and NH3, but elsewhere biomass burning is mentioned with different single-scattering albedo.

Page 1, line 6: "Almost all scenarios": Could mention here which one doesn't show a decrease.

Page 1, lines 11-13: The different SSP scenarios do not reflect an uncertainty, but rather societal choices, i.e. it is not a random outcome, but up to us to choose whether we reduce aerosols or not (there are of course uncertainties between the IAMs within each societal choice, but these aren't considered in this paper). The ERFs for the different scenarios therefore shouldn't be referred to as a spread but rather listed individually: "-0.15 for SSP1-1.9, -0.54 for SSP3-ref". To make the abstract clearer I suggest to only list ERFs, as readers can easily find the RFs in the text if they want.

Page 1, lines 13-14: Similarly the uncertainties in physics shouldn't be mixed with choices in scenario. Rather list the effects on the two extreme scenarios: "uncertainty in Twomey effect could increase these to -0.39 and -0.92.".

Page 1: There should also be a statement of the aerosol forcings provided by the IAMs themselves (-0.365 for SSP1-1.9, -1.017 for SSP3-ref).

Page 2, line 18: There should be some comment here or later in the text about how reasonable the linear relation between tau and emissions is.

Section 2.2: This methodology wasn't easy to follow. If the sum in eqn (2) is simply over SO2 and NH3 this should be make more explicit. Does the statement that these source "contribute one third of the total global emissions in 2005" refer to total anthopogenic+natural? If it is anthropogenic only, where do the other two thirds come from? Is the SO2/NH3 weighting the same for open burning as industrial sources – and if so why?

Section 2.3: This methodology wasn't easy to follow. Some tables listing all the experiments would help. It is not clear whether some of the multiples of three are identical ensembles or whether they have varying parameters (such as eta_N). How does this then lead to 180 annual estimates of ERF and thirty estimates of RF?

Page 5, line 14: LBG should be spelled out in full, along with a short description of the implication of lowering the background.

Section 3. This needs to be clearer about which experiments are run with the 9 scenarios in table 1 and which with the 3 scenarios in table 2.

Page 7, lines 31-4: The RCP scenarios used by studies cited here had very different aerosol emissions to the SSPs. Does it make sense to say your estimates are consistent?

Page 8, lines 8-9. It is not necessarily the variability in the rapid adjustments that causes the variability in the ERF, but rather that the methodology used is sensitive to the interannual variability in the clouds. For instance the ERF for no change in aerosols will still have a large interannual variability even though we know the rapid adjustment (and ERF) is exactly zero in every year (and indeed every timestep).

Page 8, line 12: "Efficacy" is often used for temperature response. Suggest to use "efficiency" here.

Page 8, lines 14-15: It is not obvious how the authors know the edge of the biomass burning plume is more strongly absorbing as the optical depth is only based on SO2 and NH3 emissions?

Page 8, lines 22-24: This apparent positive rapid adjustment needs further discussion. Known adjustment processes tend to be negative (e.g. Smith et al. 2018 https://doi.org/10.1029/2018GL079826 ). These apparent adjustments could be the effect of circulation changes on the clouds.

Page 9, lines 7: As with the abstract these numbers should not be referred to as a spread, but as ERFs for the SSP1-1.9 and SSP3-ref scenarios.

Page 9, lines 10-11: As before, it does not make sense to add in physical uncertainties to difference in scenario choice.

Figure 1: It would be better to keep the same scale for all these.

Figure 6: Use "Efficiency" rather than simply "E".

[Figure]

---

## Author Comment (AC1) · 14 Jan 2019

**Reply to the reviewers for the manuscript:**
**"**First forcing estimates from the future CMIP6 scenarios of anthropogenic aerosol optical properties and an associated Twomey effect" by Fiedler et al.

We would like to thank William Collins and the anonymous reviewer for their comments that helped improving our manuscript. In the following, we give our reply (blue) to each of the reviewer's comments (black) and document the associated changes in the manuscript.

**Reviewer #1**
Fiedler et al. present a modelling study in which they interpret the future emission scenarios of Riahi et al. (2017) using a simple model implemented in a GCM. The two aspects that provide added value compared to the Riahi et al. paper in my opinion are that geographical distributions are shown here, and that the scaling of Stevens et al. (2017) allows to convert the emissions into forcing values given the assumptions in the simple MACv2-SP approach (with some extra model information added from the simulated cloud fraction- and cloud droplet concentration distributions). As far as I understand, one of the co-authors, Gidden, prepares another manuscript for Geophys. Model Devel. that possibly covers the former aspect in a similar way.

Thank you for your comments. Matt Gidden, a co-author of this study, prepared another manuscript while we have developed our joint article. We cite his article in our article and update the reference and the names of the scenarios from his article in our revised manuscript. Our content substantially differs from that study. We here document the MACv2-SP interpretation of the CMIP6 scenarios and show the first forcing estimates for anthropogenic aerosol of the MACv2-SP interpretation of the CMIP6 emission scenarios.

The manuscript suffers from poor explanations. It is very difficult to follow the set-up of the simulations (e.g. number of years integrated, boundary conditions for these simulations).

We kept the description of the simulations themselves short and referred to the detailed description documented elsewhere (Section 2.3). In the revised manuscript, we include more information in Section 2.3. Please refer to our detailed replies below.

The main problem, however, is that it is completely unclear which aerosol species are assessed. In subsection 2.1 of the "Methods" section (p3, l12) the authors explain they distinguish between "biomass burning" and "industrial" aerosol emissions, with substantially different single scattering albedo. This interpretation is taken up later in the discussion of results. In contrast, subsection 2.2 (p4, l7) explains that only sulfate and nitrate are investigated. In summary, after these explanations are added, and after the specific comments are addressed, I believe the paper has some merit and should be published in Geosci. Model Devel.

The anthropogenic aerosol optical properties in MACv2-SP is for all anthropogenic aerosol species. We choose a different single scattering albedo for industrially polluted regions and regions additionally affected by biomass burning for accounting for regional differences in aerosol absorption (Section 2.1). We add in Section 2.2: "Following the approach by Stevens et al. (2017), we assume that the emission of all anthropogenic aerosol species scale with the emission of $SO_2$ and $NH_3$, and therefore use these two species for scaling the anthropogenic aerosol optical depth of MACv2-SP over time. We herein use $NH_3$ emissions in addition to $SO_2$ for considering that not all dominant aerosol emission changes over time scale with the $SO_2$ development. (...) Aerosol absorption is represented by the single scattering albedo (Section 2.1)"

p1 l12 – it might be instructive to also report the 2090s forcing relative to 2015.
Compiling the historical evolution of the aerosol forcing is planned for RFMIP and has not yet been carried out in the framework of this study. We add in the discussion section: "RFMIP (...) will provide aerosol forcing estimates from 1850 to 2100"

p2 l10 – Mauritsen et al.: drop "and"
Dropped.

p3 l5 – "need" ! "desire"
Replaced with "desire"

p3 l27 – w is not explained. Is it some sort of weight? I don't think it is meaningful not to specify the number of species in the sum (and in the text). Isn't it just two?
In the revised manuscript, we move the explanation of the weights $\omega_k$ on page 4 l 5-7 to the previous paragraph: "The weight $w_k$ describes the relative contribution of the two species, namely $w_1 = 0.645$ for SO2 and $w_2 = 0.355$ for NH3, motivated by the present-day ratio between sulphate and ammonia forcing as in Stevens et al. (2017)." and explicitly name the numbers ("k=1,2") in the equation.

p3 l29 – p4 l1: I do not understand what the authors do here or mean. A figure might be helpful, or more text to explain.
We add a new figure for illustrating the plume centres in MACv2-SP and change here in the text: "(…) in a twenty by twenty degree box around each plume centre, marked in Figure 1.". We further add in Section 2.1: "Figure 1 shows the annual mean of the aerosol optical depth of MACv2-SP for 2005 and the location of the nine plume centres for constructing the spatial distribution."

p4 l4 – is that result shown somewhere, or is there a reference? - the statement needs to be corroborated or withdrawn.
We now add a figure on the comparison in the appendix and refer to it in the text: "We test the reproducibility of the regional evolution of $\tau_i$ by scaling with emissions averaged around the plume centers. For doing so, we derive $E_i$ from a pre-existing aerosol emission database adopting the same spatial averaging, and compare the results to the corresponding $E_i$ directly derived from the aerosol optical depth in a simulation with the aerosol-climate model ECHAM-HAM that uses the same aerosol emissions as boundary data (Figure A1). Using spatial averages of aerosol emissions around the plume centres gives $E_i$ similar to the direct scaling from the time-evolving aerosol optical depth from the complex model. The results for $E_i$ are herein not strongly sensitive to the choice of the number of grid boxes, e.g. a box of ten by ten degree around the plume centres only weakly modifies $E_i$ in most cases."

p4 l8 – the relative relevance of sulfate and nitrate depends on the abundance of SO2 and oxidants. It thus seems an oversimplification to assume it is constant with time and geographical location. Some assessment of the error introduced is at least necessary.
We show the comparison of using emission scaling and a complex aerosol-climate model for calculating the aerosol optical depth. See aloft. Here, we add: "The approach is a simplification and is meant for facilitating experimentation and a better understanding of model errors (Stevens et al., 2017)."

p4 l9 – some discussion why in particular absorbing aerosol can be neglected is necessary.
We do not neglect absorbing aerosol, but assume that the burden of absorbing aerosol scales with $SO_2$ and $NH_3$. Regional differences in absorption are represented by the single scattering albedos following Stevens et al. (2017). In addition to the details in Section 2.1., we add here: "Aerosol absorption is represented by the single scattering albedo (Section 2.1)"

p4 l12 – is this the global mean value (if so, only 0.5% difference?)? Is this similar in individual regions?

We add: "(…) global mean of (…). Regional differences are up to +/-0.039 with smaller values for 2015 in East Asia, Europe, and North America, and larger values in the other plumes".

p4 l26 – how can this affect clear-sky radiation balance?

Natural aerosols affect the radiation transfer, e.g., desert dust and sea spray. The atmospheric burden of natural aerosols is uncertain such that the clear-sky radiation balance differs amongst models. Moreover, cloud properties can be tuned. So, we generalise the statement in the revised manuscript: "to optionally tune the radiation balance of models".

p5 l8 – these are equilibrium simulations, if I understand correctly. Then "six" does not mean anything, but the length of the integration is relevant.

p5 l11 – similarly, what does "three experiments" mean in this context?

We add a new table for summarising the experiment setups and revise this section for clarifying the meaning: "For sufficiently accounting for the natural variability, we run ensembles of three simulations with the pre-industrial aerosol of 1850 and the anthropogenic aerosol from MACv2-SP. All simulations are performed for the period 2000-2010 with the same annually repeating monthly anthropogenic aerosol patterns, e.g., monthly means of the year 2005. The simulations use the same year-to-year changes of the boundary conditions, e.g., observed sea-surface temperatures. This approach is chosen for representing natural variability. The first year of each simulation is considered as the spin-up period and not used in the data analyses." and modify the paragraph on ERF: "For calculating the effective radiative forcing (ERF) of anthropogenic aerosol relative to pre-industrial, we perform experiments without $\tau$ of MACv2-SP. For this reference setup, we run an ensemble of six simulations for 2000-2010 without anthropogenic aerosol, but otherwise the same initial and boundary conditions as for the simulations with $\tau$ of MACv2-SP for efficiently increasing the number of estimates for ERF. ERF is determined as annual differences in the top of the atmosphere shortwave radiation balance from the three simulations with additionally $\tau$ from MACv2-SP and six simulations without $\tau$ from MACv2-SP. Since each simulation provides ten years for the analysis, we yield a total of 180 annual estimates of ERF for each anthropogenic aerosol pattern. "

p5 l13 – more precisely, this is decreasing the tunable parameter ntau_gl, I believe?

We decrease $\tau_{gl}$ for increasing $\eta_N$, and add in Section 2.1: "that increases $\eta_N$ (Equations 3 and 4) "

p5 l14 – what motivates the name "LBG"?

LBG stands for "low background" referring to the smaller value of $\tau_{gl}$ following the name of the experiment type in Fiedler et al. (2017). We add: "low background (LBG)"

p5 l17 – what defines a specific year (i.e. why call it "2000 – 2010" rather than "2005 aerosol")? Is the sea surface temperature from observations?

Changed to: "All simulations are performed for the period 2000-2010 with the same annually repeating monthly anthropogenic aerosol patterns, e.g., monthly means of the year 2005. The simulations use the same year-to-year changes of the boundary conditions, e.g., observed sea-surface temperatures. This approach is chosen for representing natural variability. "

p5 l21 – what is "without ntau", especially for the Twomey effect? Don't the authors rather mean, "with scaling factor (Eq. 2) for 2090 and with scaling factor of 0"? The "180 annual estimates" make me conclude there are 6 realisations (with whatever difference between them) of 30 years integration time each. Is this correct? The authors need to explain clearly what they did.

We add a new Table 1 summarising the experiment setup and write in Section 2.3: "(…) For calculating the effective radiative forcing (ERF) of anthropogenic aerosol relative to pre-industrial, we perform experiments without $\tau$ of MACv2-SP. For this reference setup, we run an ensemble of six simulations for 2000-2010 without anthropogenic aerosol, but otherwise the same initial and boundary conditions as for the simulations with $\tau$ of MACv2-SP for efficiently increasing the number of estimates for ERF. ERF is determined as annual differences in the top of the atmosphere shortwave radiation balance from the three simulations with additionally $\tau$ from MACv2-SP and six simulations without $\tau$ from MACv2-SP. Since each simulation provides ten years for the analysis, we yield a total of 180 annual estimates of ERF for each anthropogenic aerosol pattern."  Please also refer to our reply aloft.

p5 l23 – I am lost and cannot understand why thirty. What is different between the thirty realisations?

We have 180 years of ERF and 30 years of RF for each anthropogenic aerosol pattern. Here, we have added: "Since we have three simulations for each setup with $\tau$ of MACv2-SP, we have thirty estimates of RF for each of the anthropogenic aerosol patterns." Please also refer to our replies aloft.

p5 l25 – I strongly suggest not to overload the symbol "E" (that stands for emission scaling) but to use a different one.

We have removed the symbol and write 'efficiency' throughout the revised manuscript.

p6 l25 – it would be good to motivate this analysis. I would guess that there is no added information here compared to the emission scenarios. Maybe this section can be dropped.

We add the motivation: "We assess the regional characteristics in the scenarios by quantifying hemispheric differences in $\tau$, rather than comparing mean maps of $\tau$.", which we have not assessed before. We perceive the analysis interesting for characterising the differences of MACv2-SP's interpretation of the CMIP6 aerosol emission scenarios and keep this section in the manuscript.

p8 l12 – is this correlation not just by construction of the simple model?

We expect a correlation, but we prescribe the aerosol optical properties, not the forcing itself. The forcing depends on the aerosol optical properties, but also on other factors, e.g., the albedo of the underlying surface. As such, it is useful to analyse the spatial patterns and provide the figures as reference for other models that use MACv2-SP in the future.

p8 l14 – I did miss the introduction of absorption. Didn't p4 l7 explain that only sulfate and nitrate are used?

We use the single scattering albedo for representing absorption. We add here the reference to Section 2.1 in the manuscript.

p8 l23 – a discussion on these rapid adjustments is necessary. What exactly happens here in the model?

We add: "In our model, the radiative forcing of anthropogenic aerosol from aerosol-radiation interaction and the Twomey effect induce heating perturbations. The associated change in the air temperature affects for instance the static stability of the atmosphere and thereby the circulation and embedded clouds. Such rapid adjustments cause the difference between ERF and RF, and are here summarised as net contribution."

p12 l1 – journal is missing
We add Tellus B.

Fig. 1 – a figure showing the geographical extent of the plumes is necessary. I suggest to rather use the same scaling for all panels. A global-mean curve would be useful.
We add a new Figure 1 showing the spatial pattern of the plumes, and add in the text: "Figure 1 shows the annual mean of the aerosol optical depth of MACv2-SP for 2005 and the location of the nine plume centres for constructing the spatial distribution.", and revised former Figure 1 (now Figure 2) for having identical axes. The global mean curves for the scenarios are shown in Figure 2 (now Figure 3).

Supplementary material: I wasn't able to open the netcdf file. Is there a formatting mistake?
The file ending of the tar archive was removed during the handling of the uploaded file, but the netCDF files in the archive are ok.

**Reviewer #2 (William Collins)**

This paper provides useful documentation of the MACv2-SP aerosols that some models without their own aerosols will use for their CMIP6 simulation.

We thank you for your comments. Please note that we update the scenario names throughout the manuscript for consistency with the article by Gidden et al. (submitted).

It should be made very clear in the text that the ERFs from using these aerosols are smaller in magnitude by more than a factor of 2 than assumed by the IAMs when the scenarios were created therefore (all other factors being equal) future temperatures are likely to be less warm in these scenarios (except SSP3-ref) than expected by the IAMs, or compared to interactive aerosol models whose aerosol ERFs are closer to that of the IAMs.

We add in the conclusions: "The strength of the anthropogenic aerosol forcing has implications for the temperature development in simulations with coupled atmosphere-ocean models, e.g., a relatively weak aerosol forcing like in the standard setting of MACv2-SP likely results in a relatively stronger warming signal."

Specific comments:

Page 1, line 2: It is stated that the scenarios are based on SO2 and NH3, but elsewhere biomass burning is mentioned with different single-scattering albedo.

We choose a different single scattering albedo for industrially polluted regions and regions additionally affected by biomass burning for accounting for regional differences in aerosol absorption (Section 2.1). We add in section 2.2: "Following the approach by Stevens et al. (2017), we assume that the emission of all anthropogenic aerosol species scale with the emissions for $SO_2$ and $NH_3$, and use these two species for scaling the anthropogenic aerosol optical depth of MACv2-SP over time." In the abstract, we removed the chemical species to avoid confusion.

Page 1, line 6: "Almost all scenarios": Could mention here which one doesn't show a decrease.

Changed to: "All scenarios, except SSP3-70 and SSP4-60, show a decrease (…)"

Page 1, lines 11-13: The different SSP scenarios do not reflect an uncertainty, but rather societal choices, i.e. it is not a random outcome, but up to us to choose whether we reduce aerosols or not (there are of course uncertainties between the IAMs within each societal choice, but these aren't considered in this paper). The ERFs for the different scenarios therefore shouldn't be referred to as a spread but rather listed individually: "-0.15 for SSP1-1.9, -0.54 for SSP3-ref". To make the abstract clearer I suggest to only list ERFs, as readers can easily find the RFs in the text if they want.

We remove the values for RF and modify statements in the abstract: "We estimate the radiative forcing of anthropogenic aerosol from high- and low-end scenarios in the mid-2090s (…) The ERF of anthropogenic aerosol for the mid-2090s ranges from -0.15 $Wm^{-2}$ for SSP1-19 to -0.54 $Wm^{-2}$ for SSP3-70, i.e., the mid-2090s ERF is 30-108% of the value in the mid-2000s due to differences in the emission pathway alone"

Page 1, lines 13-14: Similarly the uncertainties in physics shouldn't be mixed with choices in scenario. Rather list the effects on the two extreme scenarios: "uncertainty in Twomey effect could increase these to -0.39 and -0.92.".

Changed to: "Assuming a stronger Twomey effect changes these ERFs to -0.39 $Wm^{-2}$ and -0.92 $Wm^{-2}$, respectively, (…)."

Page 1: There should also be a statement of the aerosol forcings provided by the IAMs themselves (-0.365 for SSP1-1.9, -1.017 for SSP3-ref).

We add to the previous sentence: "(…) which are similar to estimates obtained from models with complex aerosol parameterisations."

Page 2, line 18: There should be some comment here or later in the text about how reasonable the linear relation between tau and emissions is.

We add the figure and description of the analysis on this topic in Section 2.2: "We test the reproducibility of the regional evolution of $\tau_i$ by scaling with emissions averaged around the plume centers. For doing so, we derive $E_i$ from a pre-existing database adopting the same spatial averaging, and compare the results to the corresponding $E_i$, directly derived from the aerosol optical depth from a simulation with the aerosol-climate model ECHAM-HAM that uses the same historical aerosol emissions as boundary data (Figure A1). Using spatial averages of aerosol emissions around the plume centres gives $E_i$ similar to the direct scaling from the time-evolving aerosol optical depth from the complex model. The results for $E_i$ are herein not strongly sensitive to the choice of the number of grid boxes, e.g. a box of ten by ten degree around the plume centres only weakly modifies $E_i$ in most cases."

Section 2.2: This methodology wasn't easy to follow. If the sum in eqn (2) is simply over SO2 and NH3 this should be make more explicit. Does the statement that these source "contribute one third of the total global emissions in 2005" refer to total anthopogenic+natural? If it is anthropogenic only, where do the other two thirds come from? Is the SO2/NH3 weighting the same for open burning as industrial sources – and if so why?

We revise this section, and herein use also the comments of the first reviewer. The revised manuscript has the explanation of the weights $\omega_k$ on page 4 l 5-7 in the previous paragraph: " The weight w$_k$ describes the relative contribution of the two species, namely w$_1$ = 0.645 for SO2 and w$_2$ = 0.355 for NH3 , motivated by the present-day ratio between sulphate and ammonia forcing as in Stevens et al. (2017)." and explicitly names the numbers : "k=1,2" in the equation. We further clarify: "(…), i.e., these regions capture the dominant anthropogenic sources and contribute one third of the total global anthropogenic emissions in 2005." The SO2/NH3 weighting is the same for all plumes. We add: "The approach is a simplification and is meant for facilitating experimentation and a better understanding of model errors (Stevens et al., 2016).". Please also refer to our replies to reviewer #1 for changes in this section or the manuscript with highlighted changes for more details on the revision of this Section.

Section 2.3: This methodology wasn't easy to follow. Some tables listing all the experiments would help. It is not clear whether some of the multiples of three are identical ensembles or whether they have varying parameters (such as eta_N). How does this then lead to 180 annual estimates of ERF and thirty estimates of RF?

We add a new Table 1 for an overview on the experiments, added: "The setup of the experiments are summarised in Table 1. ", and revised Section 2.3 for clarity: Since we have three simulations for each setup with $\tau$ of MACv2-SP, we have thirty estimates of RF for each of the anthropogenic aerosol patterns. (…) For calculating the effective radiative forcing (ERF) of anthropogenic aerosol relative to pre-industrial, we perform experiments without $\tau$ of MACv2-SP. For this reference setup, we run an ensemble of six simulations for 2000-2010 without anthropogenic aerosol, but otherwise the same initial and boundary conditions as for the simulations with $\tau$ of MACv2-SP, for efficiently increasing the number of estimates for ERF. ERF is determined as annual differences in the top of the atmosphere shortwave radiation balance from the three simulations with additionally $\tau$ from MACv2-SP and six simulations without $\tau$ from MACv2-SP. Since each simulation provides ten years for the analysis, we yield a total of 180 annual estimates of ERF for each anthropogenic aerosol pattern."

Page 5, line 14: LBG should be spelled out in full, along with a short description of the implication of lowering the background.
We add: "Here, we follow the method of the low background (LBG) experiments in Fiedler et al. (2017) and set $\tau_{gl} = 0.002$ that increases $\eta_N$ (Equations 3 and 4) in the experiments SSP1-26-LBG, SSP3-70-LBG, and SSP5-85-LBG of the present article."

Section 3. This needs to be clearer about which experiments are run with the 9 scenarios in table 1 and which with the 3 scenarios in table 2.
We change the introduction of Section 3.2: "We choose three scenarios for assessing the differences in the radiative forcing of anthropogenic aerosol in the mid-2090s associated with the choice of the emission pathway (Table 1). These are SSP3-70 as high-end scenario and SSP1-26 as a lower bound for the $\overline{\tau}$ spread of 0.009 to 0.027 at the end of the 21st century. The third scenario choice is SSP5-85 (…)"

Page 7, lines 31-4: The RCP scenarios used by studies cited here had very different aerosol emissions to the SSPs. Does it make sense to say your estimates are consistent?
The forcing values are similar. We change: "estimates" to "forcing values" for clarity.

Page 8, lines 8-9. It is not necessarily the variability in the rapid adjustments that causes the variability in the ERF, but rather that the methodology used is sensitive to the interannual variability in the clouds. For instance the ERF for no change in aerosols will still have a large interannual variability even though we know the rapid adjustment (and ERF) is exactly zero in every year (and indeed every timestep).
We add an explanation of the rapid adjustments in our model: "In our model, the radiative forcing of anthropogenic aerosol from aerosol-radiation interaction and the Twomey effect induce heating perturbations. The associated change in the air temperature affects for instance the static stability of the atmosphere and thereby the circulation and embedded clouds. Such rapid adjustments cause the difference between ERF and RF, and are here summarised as net contribution.". Our definition of ERF, RF and the net contribution of rapid adjustments is given in Section 2.3.

Page 8, line 12: "Efficacy" is often used for temperature response. Suggest to use "efficiency" here.
Replaced with "efficiency".

Page 8, lines 14-15: It is not obvious how the authors know the edge of the biomass burning plume is more strongly absorbing as the optical depth is only based on SO2 and NH3 emissions?
The optical properties of all anthropogenic aerosol for the mid-2000s are from the climatology MACv2 (Section 2.1), and we use SO2 and NH3 for the temporal scaling into the future (Section 2.2). We prescribe single scattering albedos for representing aerosol absorption. The single scattering albedo is smaller in biomass burning regions than in the plumes for industrial pollution, thus the absorption of the biomass burning aerosol is larger. We mark the biomass burning plumes in the new Figure 1 and add here the reference to Section 2.1 for details on the single scattering albedo.

Page 8, lines 22-24: This apparent positive rapid adjustment needs further discussion. Known adjustment processes tend to be negative (e.g. Smith et al. 2018 https://doi.org/ 10.1029/2018GL079826 ). These apparent adjustments could be the effect of circulation changes on the clouds.
Yes, we add the explanation of the "rapid adjustments in the atmosphere" of our model at the end of the paragraph. Please refer to our reply aloft for the changes in the manuscript.

Page 9, lines 7: As with the abstract these numbers should not be referred to as a spread, but as ERFs for the SSP1-1.9 and SSP3-ref scenarios.

We change these expressions throughout the manuscript. Here we change it to: "We estimate the differences in the radiative forcing of anthropogenic aerosol at the end of the 21st century that are associated with the choice of the future aerosol emission scenario (Fig. 11). For doing so, we choose three aerosol forcing scenarios that include the high- and low-end scenarios of $\tau$ in the mid-2090s. (…) MPI-ESM1.2 gives -0.15 Wm$^{-2}$ with SSP1-26 to -0.54 Wm$^{-2}$ with SSP3-70 for the ERF of anthropogenic aerosol for the mid-2090s (Fig. 11), reflecting the overall differences due to the anthropogenic emission pathways alone. The clear-sky forcing is herein slightly stronger with -0.24 Wm$^{-2}$ to -0.69 Wm$^{-2}$, respectively, since the clouds mask radiative effects of anthropogenic aerosol." We use 'difference' instead of 'uncertainty' in the revised Figure 11 (former Figure 10).

Page 9, lines 10-11: As before, it does not make sense to add in physical uncertainties to difference in scenario choice.

Changed to: "Assuming a stronger Twomey effect gives more negative all-sky ERFs of -0.39 Wm$^{-2}$ to -0.92 Wm$^{-2}$." We also modified Figure 11 (former Figure 10) accordingly.

Figure 1: It would be better to keep the same scale for all these.

We modified the figure for the same axes in all subfigures.

Figure 6: Use "Efficiency" rather than simply "E".

We replace 'E' with 'efficiency' throughout the manuscript.

---

## Author Response (AR2)

**Reply to the reviewer for the manuscript:**
"First forcing estimates from the future CMIP6 scenarios of anthropogenic aerosol optical properties and an associated Twomey effect" by Fiedler et al.

We would like to thank the editor Andrea Stenke and an anonymous reviewer for their comment. Please find (blue) our reply below (black) the comment.

Reviewer #1

The authors mostly responded well to my concerns.
Thank you again for your comments.

There are two items that need to be clarified before publication:
1.  the new ECHAM-HAM study. I wonder whether this model includes nitrate (to my knowledge it doesn't). How can it be used to study the relative effects of sulfate and nitrate?
The model simulation with ECHAM-HAM did not have nitrate. The intention of showing this analysis is an assessment of the reproducibility of the development of the aerosol optical depth by scaling with anthropogenic aerosol emissions (Section 2.2). The relative contribution of nitrate and sulphate is based on Stevens et al. (2017). We add in the caption of Figure A1: „(…) using a pre-existing aerosol emission database (…)".

2. I still cannot read the supplementary netcdf file.
I have been in contact with the journal for asking for a more intuitive solution for providing the data in the supplementary material. The journal personnel would intend to revise the file format in the course of the revision of the manuscript. For the time being, please add „.tar" to the file in the folder after downloading the supplementary material and open the archive accordingly. You will then see the folder containing the forcing files in netCDF format and a Readme file.

[Figure]

**Figure A1.** Scaling factor comparison. Shown are annual scaling factors $E_i(t)$ derived from (black) the aerosol optical depth in the plume centres of a transient ECHAM-HAM simulation using a pre-existing aerosol emission database, and (colours) the anthropogenic aerosol emissions of that simulation, averaged over grid boxes around the plume centres. The geographical positions of the plumes with (circles) industrial pollution and (rectangles) biomass burning are indicated.